# Three-Dimensional Bioprinted Skin Microrelief and Its Role in Skin Aging

**DOI:** 10.3390/biomimetics9060366

**Published:** 2024-06-17

**Authors:** Wenxuan Sun, Bo Wang, Tianhao Yang, Ruixue Yin, Feifei Wang, Hongbo Zhang, Wenjun Zhang

**Affiliations:** 1School of Mechanical and Power Engineering, East China University of Science and Technology, Shanghai 200237, China; swx_1204@126.com (W.S.); yangth1998@163.com (T.Y.); yinruixue@ecust.edu.cn (R.Y.); 2Yunnan Botanee Bio-Technology Group Co., Ltd., Kunming 650033, China; wbsusie@163.com; 3Yunnan Characteristic Plant Extraction Laboratory, Yunnan Yunke Characteristic Plant Extraction Laboratory Co., Ltd., Kunming 650106, China; 4Division of Biomedical Engineering, University of Saskatchewan, Saskatoon, SK S7N 5C9, Canada; chris.zhang@usask.ca

**Keywords:** skin wrinkles, microrelief, skin aging, DLP, Voronoi algorithm

## Abstract

Skin aging is a complex physiological process, in which cells and the extracellular matrix (ECM) interreact, which leads to a change in the mechanical properties of skin, which in turn affects the cell secretion and ECM deposition. The natural skin microrelief that exists from birth has rarely been taken into account when evaluating skin aging, apart from the common knowledge that microreliefs might serve as the starting point or initialize micro-wrinkles. In fact, microrelief itself also changes with aging. Does the microrelief have other, better uses? In this paper, owing to the fast-developing 3D printing technology, skin wrinkles with microrelief of different age groups were successfully manufactured using the Digital light processing (DLP) technology. The mechanical properties of skin samples with and without microrelief were tested. It was found that microrelief has a big impact on the elastic modulus of skin samples. In order to explore the role of microrelief in skin aging, the wrinkle formation was numerically analyzed. The microrelief models of different age groups were created using the modified Voronoi algorithm for the first time, which offers fast and flexible mesh formation. We found that skin microrelief plays an important role in regulating the modulus of the epidermis, which is the dominant factor in wrinkle formation. The wrinkle length and depth were also analyzed numerically for the first time, owing to the additional dimension offered by microrelief. The results showed that wrinkles are mainly caused by the modulus change of the epidermis in the aging process, and compared with the dermis, the hypodermis is irrelevant to wrinkling. Hereby, we developed a hypothesis that microrelief makes the skin adaptive to the mechanical property changes from aging by adjusting its shape and size. The native-like skin samples with microrelief might shed a light on the mechanism of wrinkling and also help with understanding the complex physiological processes associated with human skin.

## 1. Introduction

Skin wrinkles are the primary sign of skin aging and are formed under the action of external factors such as aging or excessive exposure to sunlight. Owing to improved living standards, there has been an increasing demand for anti-aging products. In particular, skincare and rejuvenation products drive the market, with a compound annual growth rate of 5.7% in 2018, and they are expected to reach USD 66.2B by 2023, of which anti-wrinkle products account for the largest segment. The anti-aging market for anti-wrinkle products was valued at USD 8400.0 Mn in 2018 and will reach USD 13,940.3 Mn by 2027 [1].

The skin has a natural surface topography called the microrelief, consisting of grooves and ridges, also known as epidermal grooves, or zigzag patterns that cross each other to form a polygonal platform of rectangles, squares, trapezoids, and triangles [2]. Microreliefs are of great importance not only for wrinkling, but also for mimicking sensations, which is also an important aspect of wearable product development. The natural skin surface is dimpled in some places and raised in others, along with collagen. With increasing age, collagen and elastin fibers are arranged in a certain direction, and the polygon pattern loses its isotropic distribution and becomes more anisotropic by forming the preferred structural orientation [3], forming the primary and secondary lines. These main lines help control sebum volume and sweat movement, and the direction of the primary and secondary lines is also related to the underlying skin anisotropy and Lange’s line [4]. As the skin ages, there is less collagen and dermal support, deeper grooves, darker microreliefs, and looser skin. The shape and size of the microrelief also change, but the causes of these changes have rarely been studied.

The mechanism of skin wrinkle formation has been intensively evaluated in recent years, and numerical simulations have been performed to evaluate the effect of the multi-layer structure of skin on the emergence of wrinkling. Juan G et al. proposed a three-dimensional numerical model to investigate the effect of shear contact force on skin wrinkles. The dependence of skin friction on the indenter’s size, the contact material, and the tissue’s stiffness was evaluated [5,6]. In addition to the effect of external mechanical stimulation, the body’s own actions and expressions to stretch and compress the skin also have an impact on the production of wrinkles. Lejeune et al. [7] developed an algorithm to study multi-layer wrinkling. Zhao et al. [8] further proposed a six-layered skin model that considered the role of the dermal–epidermal junction (DEJ) and predicted the formation of dynamic and static wrinkles in the forehead. These results suggest that there were three wrinkling modes in the forehead, where the deepest wrinkles could reach the reticular dermis. However, the microrelief that changes over the course of aging has rarely been considered, apart from when the ratio of young modules between different layers is larger than 100, which hampers wrinkling [9]. Thus, only the depth of wrinkling has been analyzed, and the length of the wrinkle has not yet been evaluated. In fact, microrelief might act as a defect in wrinkling, but it also plays an important role in regulating the skin’s physio-biological microenvironment, resulting in different mechanical properties of skin. 

Artificial skin with a tactile sensation of human-like skin has been fast developing. However, such skin structures are still not sufficiently specific; for example, the lack of skin microrelief results in less tactile sensation and inadequate functions [10]. Currently, there are two ways to obtain a fine image of the skin’s microrelief. One is to use a certain angle of light to illuminate the skin surface and then capture an image of the skin surface to evaluate or calculate the degree of skin wrinkles and microrelief according to the shadow area cast by the skin microrelief in light [11,12]. The other method is silica gel replication. In this method, the silica gel is applied to the selected test area, and the skin microreliefs of the test area are copied to the silica gel after solidification, and then, the surface microreliefs of the silica gel are photographed and analyzed. These methods have good resolution in larger microreliefs, but are not ideal for the fine skin lines [13,14]. On the one hand, owing to the small concave and convex fine skin lines, the shadow generated in the image would be difficult to detect, especially when the shooting pixel is low and difficult to distinguish. Similarly, it is difficult to completely replicate fine lines during silica gel replication. 

Digital optical projection (DLP) is a newly developed technology in recent years. Three-dimensional structures are fabricated in a layer-by-layer fashion with photo-crosslinked bio-inks [15,16]. DLP has several advantages over extrusion-based 3D bioprinting, one of which is that hollow structures, such as vascularization networks, can easily be obtained. The printed products have a smaller stepped effect and can produce fine structures, providing a good surface finish [17]. The setup time is faster than other line-by-line printers (extrusion, injection, laser micro-stereoscopic imaging). The printing time of each layer is the same, regardless of its complexity or size, as it only depends on the thickness of the structure, and the accuracy of the printing is also greatly improved compared with other methods. Therefore, it is suitable for printing skin microreliefs with a high resolution.

In this paper, the modified Voronoi algorithm was used to create microrelief samples with aging characteristics to analyze the depth and length of wrinkles; the effects of the skin layer and morphology changes were also taken into account in the analysis of wrinkle formation for the first time. DLP 3D printing technology was applied to fabricate skin samples with microrelief using hydrogels for the first time, and the role of microrelief on skin aging was evaluated.

We demonstrated that microrelief mainly affects the elastic modulus of the epidermis, which is the dominant factor in skin wrinkling, which has not yet been reported in the literature. We also developed a hypothesis that the skin topology change is compliant with the mechanical property changes from aging owing to the existence of microrelief. The details of wrinkling, such as the depth and length, were determined under different stretching conditions, mimicking the tension in different parts of the body. 

## 2. Methods

### 2.1. Materials

Gelatin Methacryloyl (GelMA) was synthesized according to the literature [18]. Briefly, 1 g of gelatin was dissolved in 50 mL of deionized water in a 50° coil bath, and 2 mL of methacrylic anhydride (MA) was added to obtain a mixed solution with a concentration of 20% (*v*/*v*). The reaction was carried out at 300 rpm for 12 h, and then 20 mL of phosphate-buffered saline (PBS) was added for dilution. The solution was then placed in a centrifuge at 7000 rpm for 10 min to remove unreacted precipitates and dialyzed for one week to prepare for use. In the PBS solution, 7.5% wt GelMA in PBS solution was obtained, and then, 1‰ of the lithiumphenyl-2, 4, 6-trimethylbenzoyl phosphinate (LAP) initiator and 5‰ of the light absorber were added to form the printing ink.

### 2.2. The Creation of Microrelief Skin Models from the Images of Human Skin of Different Age Groups

The skin model was constructed using the 3D modeling software Zbrush 2023 (Maxon Ltd., Lenexa, KS, USA) from human skin images of different age groups, in which the maximum angle, angle number, distance, and microrelief depth were used as the characteristic parameters of the skin model. The angle difference, angle number, and distance were determined from the actual images, and the microrelief depth was adjusted by contrast according to the depth of the actual skin microrelief of the human body [19,20]. Owing to the influence of uneven light irradiation, oil on the skin surface, and other factors, most of the obtained skin image samples had uneven brightness and blurred local areas. Therefore, the image was preprocessed with PHOTOSHOP to improve the image quality and reduce the irrelevant details of the target. First, the image was screened to remove blurry and incorrect parts. Then, the image was clipped to retain the center to obtain a clear skin microrelief image and converted to an 8-bit gray image to speed up the image processing. Finally, a contrast-adaptive histogram equalization process was performed, in which pixels with a high contrast or imbalance were removed to eliminate uneven brightness distributions in the image samples. After image preprocessing, the image was imported into the alpha of Zbrush and served as a microrelief model of the skin. Eight million meshes were created in total. The procedure is illustrated in Figure 1a,b. The top image of Figure 1b is the 3D model, drawn using the alpha brush, and the bottom image of Figure 1b is the 2D texture feature image after digital processing. The depth of the microrelief was determined in two steps. One was the microrelief depth of each part, determined by the contrast after the image was preprocessed. The other parameters were determined based on the overall brush depth. 

The EEMCO standard classifies skin texture according to depth and shape as follows: (1) visible wrinkles, grooves, folds, or furrows, with a depth of 100 μm to a few millimeters; (2) main lines, crossing the skin to form a parallelogram, rectangle, or square, depth of 20~100 μm; (3) secondary lines, side lines branching out from the main lines, a depth of about 5~40 μm; (4) tertiary lines, that is, small lines forming keratinocyte boundaries, with a depth of about 0.5 μm; and (5) fourth-grade lines, corresponding to the small uneven lines of keratinocytes, with a depth of about 0.05 μm [21]. In this paper, the main lines and secondary lines were constructed, and a brush with an average value of 50 μm was selected, and the brush provided depth information so that the depth of each point was different. The texture depth was measured. The deep main lines are about 80 μm, and the shallow secondary textures are about 30 μm, which meets the requirements. In addition, the characteristic parameters that characterize the skin microrelief of different age groups, such as the grid distance, maximum angle, and angle number, were strengthened in image preprocessing and calculated, as shown in Figure 1c.

### 2.3. DLP Printing of Skin Microrelief Samples

The skin microrelief models of three different age groups were created from real human skin images. Skin microrelief samples were prepared using a DLP printer, using hydrogels with precision (as shown in Figure 2). Firstly, the model was sliced, and the *stl* format of the model was created using the ShapeWare (2.1.19.27) software. In order to make the printed material not fall off, the printing time of the first layer was set to 8 s, and the base layer and ordinary layer were set to 7 s. The thickness of the printed section was selected to be 25 microns. 

### 2.4. Mechanical Testing

The sample was stretched by the tension machine (Shanghai Hengyi Precision Instrument Co., LTD, HY-0350, Shanghai, China), the prepared sample was placed in the fixture of the universal material testing machine, and the fixture was adjusted by the fine-tuning hand wheel to place it in vertical contact with the sample surface. The sample was placed under the load head so that it was in contact with the load head, and by operating the test software on the computer, it began to stretch until the sample was deformed or broken. The elastic modulus of the sample was calculated according to the load–displacement curve. When testing samples of different ages, it is necessary to set the test parameters to the same, such as the loading rate, compression degree, and temperature, so as to ensure the comparability of the test results.

The key parameters that distinguish young and aged skin’s microreliefs are the length, depth, and area of each grid (or the angle number and maximum angle) [15,22]. The samples were created using the images of human skin of different age groups as shown in Figure 3. The microreliefs could easily be identified.

### 2.5. Numerical Simulation Procedure

The skin microrelief is more of a polygon shape than triangles; furthermore, the sharp angles of triangles might cause nonconvergence in the simulation analysis. Voronoi is a continuous polygon composed of vertically dividing lines connecting two adjacent points. The generation of Voronoi is similar to lattice growth [23]. The interconnected polygon is similar to the skin microrelief, and the geometry can be adjusted quickly and easily; therefore, it was used to simulate the complex skin microrelief. However, Voronoi cannot fully simulate skin, and skin characteristics are not the same at different ages; it can be found that the microrelief of old skin is closer to Voronoi, and young skin is more triangle-like; therefore, there is a need to combine the two. In this study, Rhino 7 software (Robert McNeel Ltd., Seattle, WA, USA) embedded with the Grasshopper plug-in was used for the first time to simulate the skin microreliefs of different age groups. The specific programming of the single- and multi-layer skin models is shown in Figure 4a,b, respectively. The general idea is as follows: (1) randomly generate several points and connect the adjacent points to form a number of triangle microrelief features; (2) this is because the older the skin is, the fewer triangles will randomly merge into polygons according to the age; (3) pull all polygons into a convex and complex microrelief through a Boolean operation; (4) using a texture according to the direction of the polygon edge line, draw a few lines as a deep texture and create grooves inward to get a deeper micro-embossed structure. Figure 4c–e show a comparison of skin models generated at different ages. Figure 4f shows the four layers of the skin structure.

The detailed procedure is as follows: a 1 × 5 mm rectangle was first established, and the plane Voronoi diagram was generated with random points as the center. The convex platforms and bottom surfaces were formed using the Boolean operation as the stratum corneum (SC). Viable epidermis (VE), dermis (DE), and hypodermis (HD) layers were generated as follows: the edge lines were manually deepened to simulate primary lines. According to the literature [11,12,24], an SC layer thickness of 25 μm (primary line depth was 20 μm, secondary line was 10 μm), VE layer thickness of 60 μm, DE layer thickness of 900 μm, and HD layer thickness of 1000 μm were set, as shown in Table 1. 

The numerical simulation process was carried out as follows:(1)Material properties

A Poisson’s ratio of 0.495 was chosen for all skin layers according to Yusuke’s work [25], and as the Young’s modulus of the epidermis increased with aging, the elastic modulus of the DE did not change significantly. Therefore, the Young’s moduli of the SC and VE were varied in this study. In addition, according to previous studies [26], the HD also influences the formation of wrinkles, and the change in the Young’s modulus of the HD was tested.
(2)Grid division

The model consists of four layers. Due to the complexity of the model, with texture in the uppermost stratum corneum, the mesh division density needed to be increased. The remaining three layers were cuboid structures with different thicknesses and different degrees of deformation. In order to reduce the operation time and ensure the accuracy of the simulation, the number of grids was reduced as much as possible, so the grids were divided according to the four layers, and different grid sizes were set. From top to bottom, the four layers of grid sizes are 0.018, 0.03, 0.08, and 0.1mm, respectively, generating a total of 479,337 nodes and 232,352 units.
(3)Boundary conditions

To simulate muscle contraction, 4%, 8%, and 10% uniaxial compression was applied to the nodes at x = 5 mm along the x direction (displacement in the x direction was 0.2 mm, 0.4 mm, and 0.5 mm) to simulate the muscle contraction. To fix the entire model, the constrained x-axis displacement was 0 at x = 0 mm, the constrained x-axis displacement was 0 at y = 0 mm, and the constrained x-axis displacement was 0 at z = 0 mm. To prevent high distortion between the layers in the y direction, the rotation of the three directions was restricted to y = 1.
(4)compression computation

Wrinkle instability is a highly non-linear phenomenon. Using a standard Newton–Raphson incremental solver can result in errors [2]. Therefore, a modified Newton–Raphson incremental solution was used, and a large-deformation analysis was performed. In the solution process, there is often no convergence; therefore, the displacement convergence method was adopted, and the mesh was subdivided as much as possible to increase the number of solution steps.

## 3. Results

### 3.1. Experimental Test and Numerical Simulation of Microrelief Effects on Skin’s Mechanical Properties

Three groups of skin samples with microrelief (young, middle-aged, and elderly groups) were stretched. The tensile stress–strain curve is shown in Figure 5a. The corresponding tensile modulus of the young microrelief was 451.7 ± 29.6 kPa. The corresponding tensile modulus of the middle-aged microrelief was 487.3 ± 34.1 kPa, and that of the old microrelief was 576.0 ± 72.3 kPa. It can be seen that the tensile modulus of the old skin’s microrelief was the highest, whereas that of the young microrelief was the lowest. Table 2 shows the numerically simulated stretching results of the corresponding young, middle-aged, and old skins, and the trend is consistent with the experimental results.

It is notable that the material of the samples was the same; in other words, the Young’s moduli of the three samples were the same. However, the tensile moduli were different owing to the existence of the microrelief, which makes the skin more adaptive, like a pleat skirt, and compliant with the change in mechanical properties of the skin layers with aging. The older the skin is, the higher Young’s modulus is, and the flatter and larger the grid of the microrelief is.

To study the effects of microrelief on the elastic modulus of the skin, the same stress (0.02 MPa) was applied in the x direction, whereas Young’s modulus was set to be the same, and the strain produced was found to be significantly different. Figure 5c–e show the stretching of young, middle-aged, and old skin microreliefs, respectively. Figure 5f–h show the stress distribution when the skin was stretched for the different ages, and the comparison shows that the stress was concentrated along deep wrinkles. In addition, it was found that the greater the number of intersections connecting the skin lines is, the greater the stress is. Therefore, the maximum stress in the samples is located at the intersection of the largest skin connection line along the deep wrinkled lines. These results indicate that aged skin microrelief facilitates stress concentration, which in turn accelerates wrinkle deepening. 

### 3.2. Numerical Simulation of Wrinkling


(1)Primary microrelief of wrinkling


Skin models with and without microreliefs were created to study the effects of microrelief on wrinkle formation. To ensure that the thickness and mechanical properties of each layer of the two models were the same, the x-direction compression was set to 2%. The z-direction displacement is shown in Figure 6a. Figure 6(ai) shows wrinkling without microrelief, and Figure 6(aii) shows wrinkling with microrelief. It can be observed that the wrinkle distribution of the model without microrelief was more uniform, and there were more wrinkles. This result was consistent with the numerical simulation of Lejeune et al. [7]. The distribution of wrinkles in the model with microrelief was significantly affected by the primary lines, and most of the wrinkles deepened along the primary line. In addition, wrinkles of a shallow depth were formed between adjacent primary lines, which were almost invisible compared with those along the primary line. The wrinkle depths of the two models were similar, and the wrinkle depth of the model with primary lines was slightly shallow.

To study the influence of the compression force on the wrinkles in the x direction, the compression rates of 4%, 8%, and 10% in the x direction were applied, as shown in Figure 6(bi–iii), respectively. It can be seen that most of the wrinkles were along the primary line, and with an increase in deformation, the overall depth of these wrinkles along the primary line deepened, and the length increased. At the same time, we found that when the compression rate was 4%, the depth of the wrinkles was almost the same, and as the compression rate increased, the middle wrinkles became deeper, the wrinkles on the sides became shallower, and the geometry sagged in the middle. This resulted in agreement with that of Shiihara et al. [9].
(1)Young’s modulus’s effects on wrinkling

The Young’s modulus of the skin gradually increased with aging. A numerical simulation was performed to explore the effect of the Young’s modulus of SC on skin wrinkles. The simulation parameters were imported into the model and compressed by 8% in the x direction. This is shown in Figure 6c. When the Young’s modulus of the SC is small, shallow wrinkles are generated along the primary line. With an increase in the Young’s modulus of the SC, the wrinkles gradually deepen. When a certain degree is reached, the overall geometry sinks in the middle. The wrinkles on the edges were relatively shallow, and those in the middle were deeper, which in turn caused the deep layer of the dermis to wrinkle.

The Young’s moduli of the different layers were varied to study their influence on wrinkles. The Young’s modulus of each skin layer increased with age. This is shown in Figure 6d–f. We found that with an increase in the Young’s modulus of the skin samples, the depth of wrinkles also increased, and with an increase in the elastic modulus of the DE, the depth of wrinkles decreased. Changes in the HD had no obvious effect on wrinkle formation.

Considering the Young’s modulus of the entire skin layer, the wrinkles were not identical. As shown in Figure 7, wrinkles gradually increased with increasing age. When compressed by 8%, the wrinkle depth of young skin was 51 µm, the wrinkle depth of middle-aged skin was 137 µm, and the wrinkle depth of old skin was 724 µm. This indicates that the depth of wrinkles increases with age when the skin is subjected to the same compression. Consequently, older people are more likely to have deep wrinkles when making facial expressions than younger ones.

## 4. Conclusions

To explore the role of microrelief in skin aging, a native-like skin microrelief sample was fabricated using the DLP technique for the first time. Skin samples with microrelief along the primary lines were obtained. The mechanical properties of the skin samples with microrelief were tested experimentally. It was found that the tensile modulus of the aged skin microrelief is the highest, whereas that of the young microrelief is the lowest. We developed a hypothesis that the existence of microrelief makes the skin more adaptive, like a pleat skirt, and compliant with the change in mechanical properties of the skin layers with aging.

Using the Voronoi algorithm, we numerically evaluated the effects of microrelief on skin wrinkling. We found that there were two main contributions of microrelief: (1) With an increase in the depth of wrinkles and a decrease in the number of skin blocks, the tensile modulus of the skin gradually increased. Therefore, the tensile modulus of the skin of the elderly is larger, which is in line with the experimental results. This also confirms that microrelief plays an important role in adjusting the mechanical properties of skin. It was also found that when the skin was stressed, stress concentration occurred along deep wrinkles. (2) Deep wrinkles were more likely to develop around the primary line of the microrelief, and the furrows on the primary line became more pronounced with compression. In addition, we found that with the increase in the Young’s modulus of SC and decrease in the Young’s modulus of DE, the wrinkles deepened, whereas the influence of the Young’s modulus of HD was not significant. The interaction between microrelief, the modulus of each layer of skin, and the skin tension might be the key to skin aging.

## Figures and Tables

**Figure 1 biomimetics-09-00366-f001:**
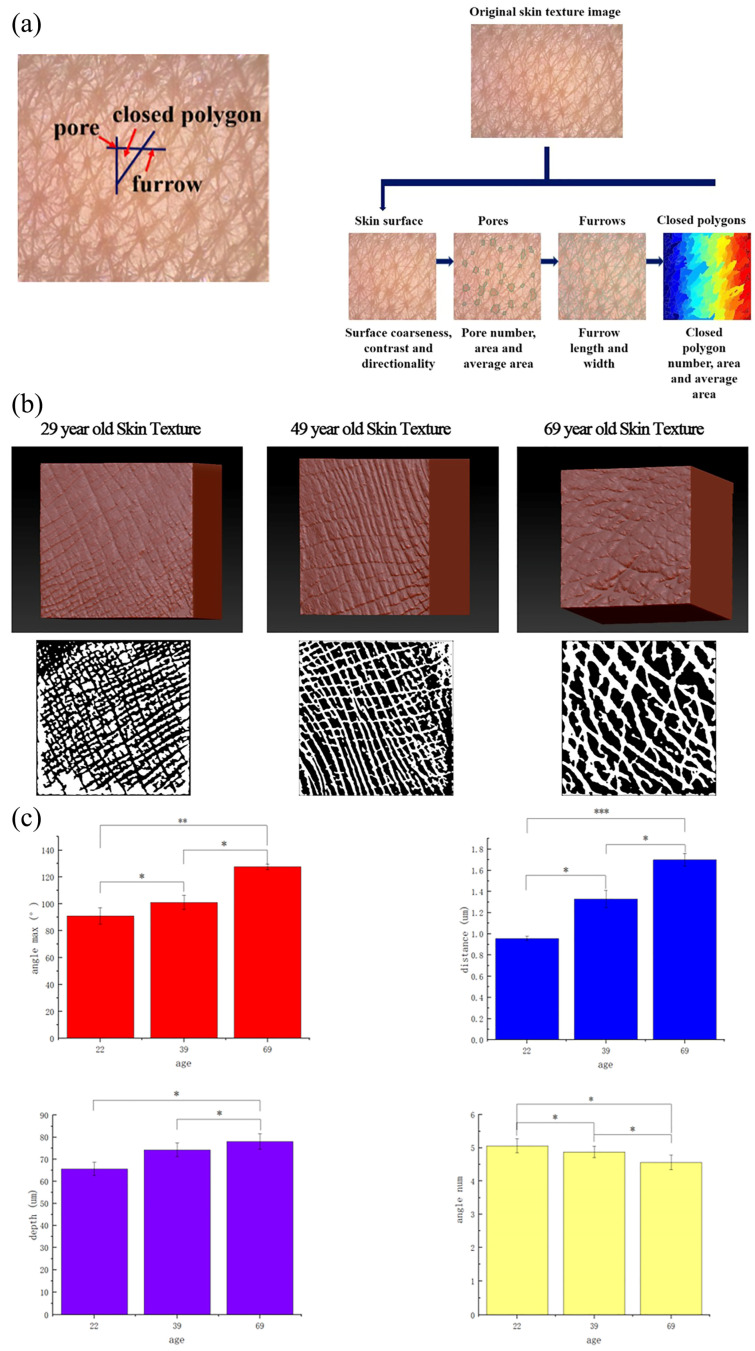
Model creation of skin sample with microrelief: (**a**) image processing [11]; (**b**) skin sample model creation; (**c**) characteristics of different age groups. The significance level is indicated as * for *p* < 0.05, ** for *p* < 0.01, and *** for *p* < 0.001, red: angle max; blue: distance; purple: depth; yellow: angle.

**Figure 2 biomimetics-09-00366-f002:**
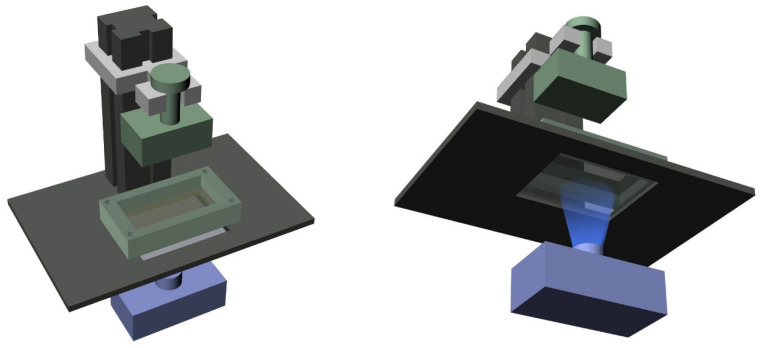
DLP fabrication of skin sample with microrelief.

**Figure 3 biomimetics-09-00366-f003:**
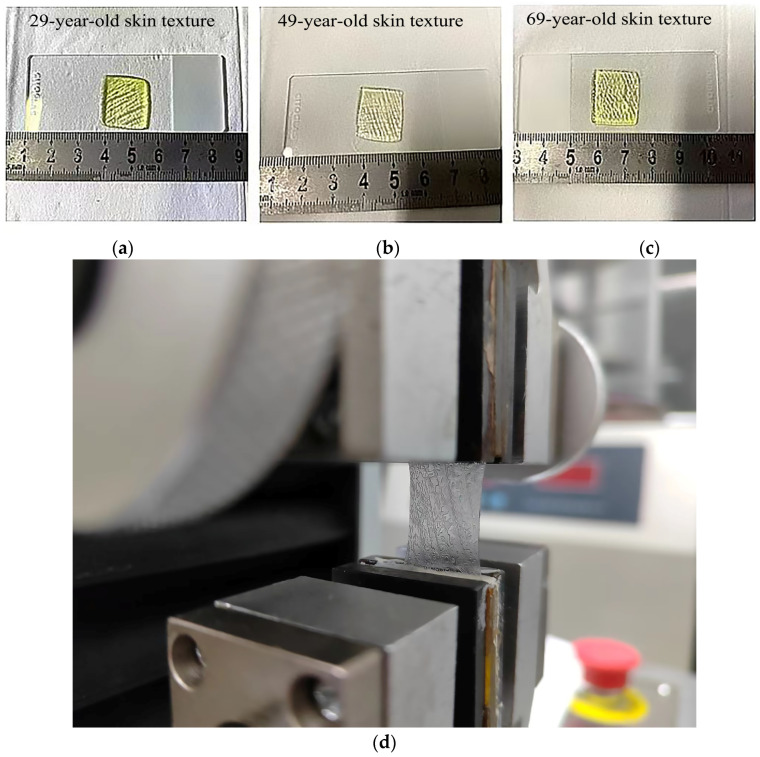
Printed hydrogel skin samples of different ages: (**a**) skin model of young person, (**b**) skin model of middle-aged person; (**c**) skin model of old person; (**d**) mechanical testing of skin microrelief samples.

**Figure 4 biomimetics-09-00366-f004:**
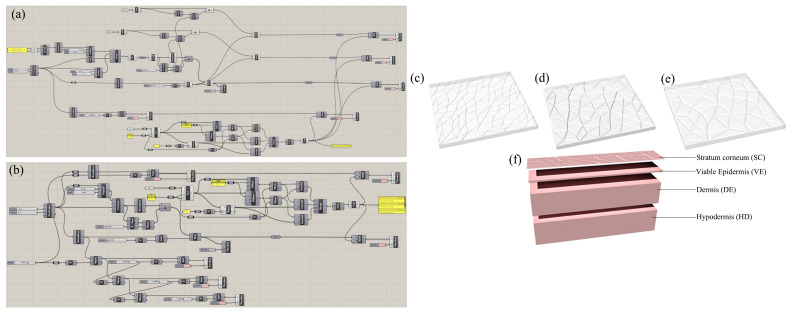
Mesh of skin samples. (**a**) Programming diagram of single-layer triangular model skin made with Grasshopper. (**b**) Programming diagram of four-layer model skin made with Grasshopper: (**c**) young skin model, (**d**) middle-aged skin model, (**e**) elderly skin model, and (**f**) four-layer skin model. Yellow: The parameter can be modified.

**Figure 5 biomimetics-09-00366-f005:**
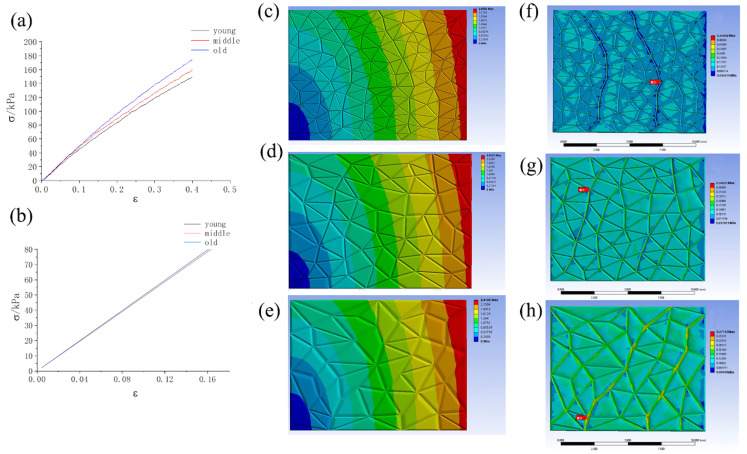
The influence of the microrelief on skin’s mechanical properties: (**a**) experimental data on stretching with different microreliefs; (**b**) simulation data of stretching with different microreliefs; (**c**–**e**) fixed Young’s modulus, application of the same stress stretching, and x direction displacement nebulae: (**c**) young, (**d**) middle-aged, and (**e**) old. (**f**–**h**) Fixed Young’s modulus, application of the same stress stretching, and the strain diagram for (**f**) young, (**g**) middle-aged, and (**h**) old.

**Figure 6 biomimetics-09-00366-f006:**
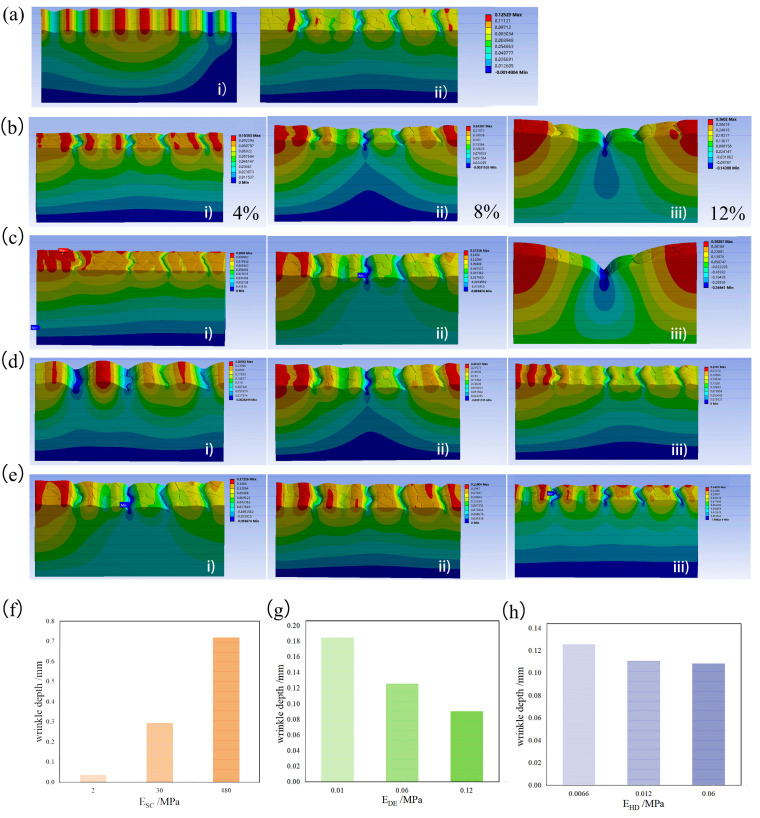
Microrelief’s effect on wrinkling (**a**) with and without microrelief: (i) no microrelief, (ii) microrelief (**b**) The displacement nephogram in the x direction of wrinkles generated by different compressions: (i) 4%, (ii) 8%, and (iii) 10%. (**c**) Effect of SC with different Young’s moduli on wrinkle depth: (i) 2 MPa, (ii) 30 MPa, and (iii) 480 MPa. (**d**) Effect of DE with different Young’s moduli on wrinkle depth: (i) 0.01 MPa, (ii) 0.06 MPa, and (iii) 0.12 MPa. (**e**) Effect of HD with different Young’s moduli on wrinkle depth: (i) 0.0066 MPa, (ii) 0.012 MPa, and (iii) 0.06 MPa. (**f**–**h**) Effect of changing the Young’s modulus of each layer on wrinkle depth. The data are the average depth of wrinkles extracted from each simulation image, and the number of repetitions is the number of wrinkles generated by the model. (**f**) SC, (**g**) DE, and (**h**) HD.

**Figure 7 biomimetics-09-00366-f007:**
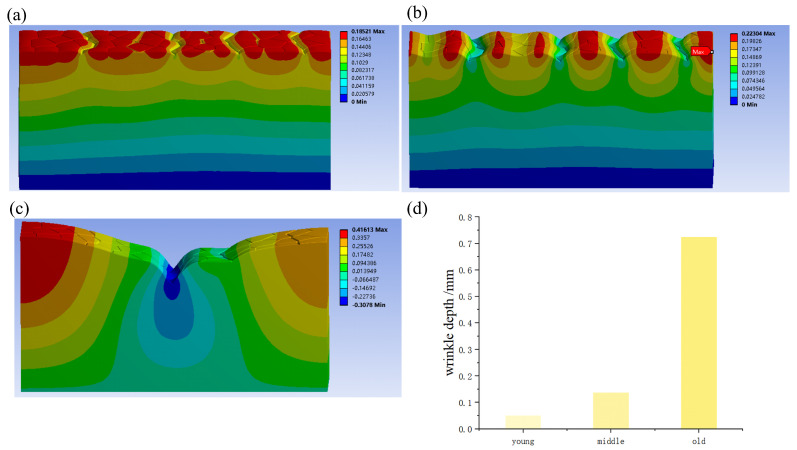
Wrinkling of entire skin. (**a**–**c**) Z-direction displacement nepicogram of young, middle-aged, and old aged skin microrelief compressed by 8%; (**d**) wrinkle depths of different age groups. The data are the average depth of wrinkles extracted from each simulation image, and the number of repetitions is the number of wrinkles generated by the model.

**Table 1 biomimetics-09-00366-t001:** The thickness and initial elastic modulus of each layer in the simulation model.

	t/mm	E/MPa
SC	0.025	2
VE	0.06	0.136
DE	0.9	0.06
HD	1	0.0066

**Table 2 biomimetics-09-00366-t002:** Comparison of experimental and simulation results for stretching microreliefs of different ages.

	Experiment	Simulation
Young	451.7 ± 29.6 kPa	491.7 kPa
Middle-aged	487.3 ± 34.1 kPa	497 kPa
Old	576.0 ± 72.3 kPa	499.1 kPa

## Data Availability

No new data were created or analyzed in this study. Data sharing is not applicable to this article.

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
