# Peer review of "Three-Dimensional Bioprinted Skin Microrelief and Its Role in Skin Aging"

_biomimetics, 2024, doi:10.3390/biomimetics9060366_

Round 1

Reviewer 1 Report

Comments and Suggestions for Authors

Overview: Problems of skin functionality increase with aging. Skin wrinkles are a cosmetic problem that partly fuel the cosmetic industry. The authors indicated that studying microrelief of skin may provide good information on skin aging causes. The authors developed a novel printed model of skin with microrelief, using pictures and software assistance. The models included epidermis, dermis, and hypodermis. A machine was used to stretch the models for comparison data. The authors found that tensile modulus increased with aging. Using math modeling, the authors found a similar conclusion, and that the hypodermis was uninvolved in wrinkle formation.

General questions to the authors about the results:

1. The uniaxial stretch testing involved some very large deformations or strains (those larger than 10%). Authors, did you track the changes of the cross-sectional areas? In large deformations, the actual deformed area is needed to accurately estimate the tensile stress.

2. The stress-strain curves are non-linear, with very little difference between different ‘age’ models for small strains (less than 0.05 or even 0.1), but bigger difference for larger strains. Authors, how did you use only one modulus to represent the mechanical properties of a non-linear and likely also viscoelastic material? 

3. The shape is not enough in any morphology-mechanics coupled structures. Although the authors captured some of the main surface microrelief shapes in their 3D printed models, it’s unclear if these models would reflect the changes in regional elasticity and residual stress (manifested by Langer lines) in aging. Authors, can you discuss these limitations?

Specific comments by the reviewer:

line 73: The authors state that the lack of skin microrelief results in less realistic feelings and inadequate functions. Authors, please cite the source of this statement. Instead of less realistic feelings, perhaps the authors mean that tactile sensation is reduced?

Line 121: How were the human skin images obtained? Did subjects provide informed consent? If so, please include this information in the methods and whether ethics/human subjects approval was granted by your institution for this study. Additionally, skin texture varies depending on the area photographed (for example, the underside of the arm is chronologically aged while the upper side demonstrates photoaging effects). From where on the person were the skin samples photographed?

In line 140, authors note that skin microrelief depth is 50 micrometers. Authors, please include a citation (at the end of the sentence) that provides experimental evidence where this measurement was derived.

On line 155-156 of the methods, the authors note that microrelief samples have high accuracy and precision. Authors, is this a result of your Figure 3 data collection? If so, please move this information to the results section, and include a graph/statistical analysis that has quantified the accuracy and precision of the models shown in Figure 3. Is this a comparison to the published studies (citations 10 and 11)? If so, please move this to the discussion section where figure 3 is discussed.

Lines 129-143 and line 169: What is the difference between the top and bottom images on Figure 1b? Authors, can you indicate in the figure legend, or in the text of lines 129-143 which descriptions and commands were used to create the images? For example, you can write “..served as a microrelief model of the skin (figure 1b, bottom images)” if this is correct. Please do the same for 1b, top images.

Line 170, figure 1 legend (C): are these bars the averages of the measured parameters, plus or minus standard deviation or standard error of the means? Please indicate in the figure legend what is being measured and number of replicates.

Line 173, Figure 3 A-C. It is difficult to see the fine details of the hydrogel skin models, even when magnifying on the screen. Please consider including a larger magnification view (or higher resolution image) of A, B, and C for the readers’ benefit.

Line 180: please include the citation again (#2 for example) that states the polygonal nature of skin microrelief.

Lines 186-187: authors, can you place the citation at the end of the statement about the triangular nature of young skin as compared to Voronoi (at the end of line 187)?

Lines 191 through 196 are difficult to understand, partly because of the presence of sentence fragments/run-ons and verb tense disagreement. Authors, please consider rewording this section for clarity. Perhaps a numbered list of commands would be better understood.

Line 283-285: is this an observation from this study or a comparison to another study? If the former, please change the verb tenses to past tense. If the latter, please include the citations.

Line 300-301, figures 6f-h, line 304, figure 7d. Are these averages? How many replicates were measured? Please include these details in the legends.

Authors, please address these 6 citations or papers in the literature not cited. These will improve your work’s impact.

1.      https://journals.plos.org/plosone/article?id=10.1371/journal.pone.0241533 please compare your results to the shear strain presented in figure 4.

2.      Line 72: The citations listed [8,9] do not describe or discuss the use of skin in robotics and artificial intelligence. Please find the correct citations or remove the sentence.

3.      Line 82: The citation listed [11] does not describe or discuss silica gel replication. Please find the correct citation (maybe #6 below?).

4, 5: Two recent papers by Juan Diosa et al. studied microrelief. Authors, how do these papers help us understand microrelief? How do their studies compare with your work?

https://asmedigitalcollection.asme.org/biomechanical/article/145/10/101008/1164223/Impact-of-Indenter-Size-and-Microrelief-Anisotropy

https://journals.plos.org/plosone/article?id=10.1371/journal.pone.0241533

6. https://pubmed.ncbi.nlm.nih.gov/27334600/  This article describes an artificial skin microrelief model silicone replica (possibly the model the authors mention in line 82?). Authors, how does it differ from your model?

Comments on the Quality of English Language

The English is good. The reviewer mentioned a couple of places where changes should be made. Section 2.4 Mechanical Testing should be converted to past tense as well because the work was done in the past.

Author Response

请参阅附件。谢谢。

Reviewer 2 Report

Comments and Suggestions for Authors

The work is interesting not only for the topic of investigation, but also for the innovative experimental approach used: 3D printing technology, the Digital light processing (DLP) technology and for the first time, microrelief models of different age groups were created using the modified Voronoi algorithm.  Three groups of skin samples with microrelief (young, middle-aged, and elderly groups) have been prepared and stretched. The tensile modulus was different owing to the existence and modification of microrelief.    which makes the skin more adaptive, with the change in mechanical properties of the skin layers with aging. It was also found that when the skin was stressed, stress concentration occurred along deep wrinkles. To explore the role of microrelief in skin aging, Paper can be improved by comparing the results obtained on the mechanical behavior of skin model prepared in this paper and the mechanical behavior of natural physiological and pathological skin obtained in bibliography   

Comments on the Quality of English Language

Minor editing of English language required

Author Response

Please see the attachment,thank you.
